

# Movement and joints: effects of overuse on anuran knee tissues

Miriam Corina Vera[1], Virginia Abdala[2], Ezequiel Aráoz[3] and María Laura Ponssa[1]

[1] Unidad Ejecutora Lillo (UEL), CONICET-Fundación Miguel Lillo, San Miguel de Tucumán, Argentina
[2] Instituto de Biodiversidad Neotropical (IBN), UNT-CONICET, San Miguel de Tucumán, Argentina
[3] Instituto de Ecología Regional, Universidad Nacional de Tucumán, Yerba Buena, Tucumán, Argentina

## ABSTRACT

Movement plays a main role in the correct development of joint tissues. In tetrapods, changes in normal movements produce alterations of such tissues during the ontogeny and in adult stages. The knee-joint is ideal for observing the influence of movement disorders, due to biomechanical properties of its components, which are involved in load transmission. We analyze the reaction of knee tissues under extreme exercise in juveniles and adults of five species of anurans with different locomotor modes. We use anurans as the case study because they undergo great mechanical stress during locomotion. We predicted that (a) knee tissues subjected to overuse will suffer a structural disorganization process; (b) adults will experience deeper morphological changes than juveniles; and (c) morphological changes will be higher in jumpers compared to walkers. To address these questions, we stimulated specimens on a treadmill belt during 2 months. We performed histological analyses of the knee of both treated and control specimens. As we expected, overuse caused structural changes in knee tissues. These alterations were gradual and higher in adults, and similar between jumpers and walkers species. This study represents a first approach to the understanding of the dynamics of anuran knee tissues during the ontogeny, and in relation to locomotion. Interestingly, the alterations found were similar to those observed in anurans subjected to reduced mobility and also to those described in joint diseases (i.e., osteoarthritis and tendinosis) in mammals, suggesting that among tetrapods, changes in movement generate similar responses in the tissues involved.

# INTRODUCTION

The mechanical load raised by movement is a key factor determining the correct morphogenesis of tetrapod joints (*Drachman & Sokoloff, 1966*; *Roddy, Prendergast & Murphy, 2011*; *Nowlan et al., 2010*; *Abdala & Ponssa, 2012*; *Shwartz, Blitz & Zelzer, 2013*; *Ponssa & Abdala, 2016*). Since the joint is a structure adapted for motion, the requirement of movement for its correct development is expected (*Drachman & Sokoloff, 1966*).

Corresponding authors
Virginia Abdala,
virginia@webmail.unt.edu.ar
María Laura Ponssa,
mlponssa@hotmail.com

Previous studies have demonstrated that alterations in the correct movement of limbs can produce severe malformations during the ontogeny, including adult pathologies (*Hosseini & Hogg, 1991*; *Arokoski et al., 2000*; *Pitsillides, 2006*; *Nowlan et al., 2010, 2012*; *Nowlan, Chandaria & Sharpe, 2014*; *Abdala & Ponssa, 2012*; *Kim et al., 2015*; *Ponssa & Abdala, 2016*; *Verbruggen et al., 2016*; *Ford et al., 2017*). The absence or reduction of movement in early stages of development produce similar phenotypical alterations both in free-living organisms, such as anurans (*Abdala & Ponssa, 2012*), or in organisms living in "controlled" environments, such as mice (*Coutinho et al., 2002*; *Kahn et al., 2009*) or chicken (*Sullivan, 1966*; *Murray & Drachman, 1969*; *Hall, 1975*; *Hall & Herring, 1990*; *Quinn et al., 1998*; *Pitsillides, 2006*). Likewise, joint-tissues that are subject to extreme mechanical loads caused by overuse can suffers similar consequences (*Shwartz, Blitz & Zelzer, 2013*) as was studied in bones and articular cartilage (*Shwartz, Blitz & Zelzer, 2013*) tendons (*Sharma & Maffulli, 2005*; *Maeda et al., 2011*), and menisci (*Adirim & Cheng, 2003*).

The knee-joint is one of the largest synovial joints in the tetrapod body, and the most vulnerable to pathologies (*Tidke & Tidke, 2013*). Each of its components, including bones, ligaments, tendons, menisci and articular cartilage, are involved in load transmission, thus the proper functioning of each structure is essential for the correct functioning of the joint (*Clark & Ogden, 1983*; *Ralphs & Benjamin, 1994*).

Tendons are pieces of connective tissue linking muscles to bones that generate the movement initiated by muscle contraction (*Kjær et al., 2006*; *Zelzer et al., 2014*). These elements are excellent biological models to study the biomechanical and morphological adaptations of connective tissues to movement (*Vilarta & De Campos Vidal, 1989*; *Feitosa, Vidal & Pimentel, 2002*). The biochemical properties of tendons and other collagenous connective tissues vary with age (*Thampatty & Wang, 2018*), and seem to be correlated with morphological changes (*Viidik, 1982*; *Shadwick, 1990*). Previous studies usually focused on the differences in tendon properties related to age (*Viidik, 1982*), and on the effects of mechanical load on tendons at different ages in rats (*Ingelmark, 1948*), mice (*Michna, 1984*), humans (*Kannus et al., 1997*) and horses (*Edwards et al., 2005*). *Ingelmark (1948)* observed that the thickness of collagen fibers did not vary in rats of different age, but found differences in younger individuals subjected to physical training. At every age, tendon cells are able to react to changes in mechanical loads, and to alter the composition of their extracellular matrix, forming a fibrocartilaginous matrix (*Benjamin & Ralphs, 1998*). The fibrocartilage usually occurs where tendons wrap and insert bones (*Benjamin, Tyers & Ralphs, 1991*; *Vogel, 2003*), and is maintained due to the mechanical stimuli acting on the joint (*Ralphs & Benjamin, 1994*; *Carvalho & Felisbino, 1999*). Although tendons show good ability to adapt to loading and movement (*Kannus et al., 1997*), if the tissue does not have time to repair itself it may not adjust, leading to injuries (*Selvanetti, Cipolla & Puddu, 1997*; *Sharma & Maffulli, 2005*). Tendon pathologies have been often studied in the human Achilles tendon (*Kader et al., 2002*; *Cook et al., 2004*; *Maffulli et al., 2008*) and are evident in animals subjected to great mechanical loads, such as physical exercises (*Kannus et al., 1997*; *Kraushaar & Nirschl, 1999*; *Thampatty & Wang, 2018*). Other structures vulnerable to age and mechanical

alteration of the joint environment are the menisci and the articular cartilages, that also play a key role in the correct functioning of the knee-joint (*Poole et al., 2001*; *Tomkoria, Patel & Mao, 2004*; *Senan et al., 2011*; *Sun et al., 2012*). The articular cartilage in conjunction with the synovial fluid provides a frictionless articulation, and absorbs and dissipates load (*Poole et al., 2001*). The properties of the articular cartilage are provided by the extracellular matrix and its chondrocytes (*Senan et al., 2011*). Within the articular cartilage, different zones can be recognized, with different properties according to the mechanical requirements (*Tomkoria, Patel & Mao, 2004*). These biomechanical properties provide perfect support for the normal movement of the synovial joints, but also make it vulnerable to extreme mobility or immobilization (*Ni et al., 2015*). The menisci also play a mechanical role as stabilizers and weight-transmitters in the knee (*Clark & Ogden, 1983*; *Senan et al., 2011*). The normal functioning of the menisci depends on their correct biochemical composition, ultrastructural organization, matrix composition and cellularity (*Senan et al., 2011*; *Pauli et al., 2011*). The number of cells in the articular cartilage and the menisci are important parameters for inferring their degree of alteration (*Tomkoria, Patel & Mao, 2004*) and the capacity to heal the tissue (*Pauli et al., 2011*). Also, both structures are functionally related; indeed, magnetic resonance images revealed that degeneration in the menisci is a potential risk factor of osteoarthritis due to its close relationship with the articular cartilage (*Sun et al., 2012*).

Different exercises have distinctive mechanical requirements (*Ebben et al., 2011*) with specific effects over the skeletal tissue (*Frost, 1994*; *Sharma & Maffulli, 2005*; *Ebben et al., 2011*). The saltatory locomotion mode of anurans is one of the most challenging among tetrapods, due to the mechanical stress raised (*Lutz & Rome, 1994*; *Peplowski & Marsh, 1997*; *Bennett, 2001*; *Nauwelaerts, Stamhuis & Aerts, 2005*; *Přikryl et al., 2009*; *Astley et al., 2013*; *Astley, Haruta & Roberts, 2015*; *Astley & Roberts, 2012*, *2014*). Although jumping is the dominant locomotion mode in anurans (*Přikryl et al., 2009*), hopping, swimming and/or walking are also present (*Emerson, 1979*; *Jorgensen & Reilly, 2013*). In these animals, locomotion has been studied from a biomechanical and anatomical perspective of the pectoral and pelvic girdles (*Emerson, 1979*; *Přikryl et al., 2009*; *Jorgensen & Reilly, 2013*; *Fabrezi et al., 2014*) and limbs (*Kargo, Nelson & Rome, 2002*; *Nauwelaerts & Aerts, 2003*, *2006*). Specific studies of the knee-joint and the dynamics of its tissue are scarce (*Hebling et al., 2014*; *Ponssa & Abdala, 2016*; *Abdala, Vera & Ponssa, 2017*) despite its important role supporting great mechanical loads. Accordingly, the knee-joint of anurans is an excellent study case to observe the dynamics of knee connective tissues (tendons, fibrocartilage and articular cartilage) subjected to intense exercise.

Here, we present new data of the effect of the mechanical stress in the anuran knee' joints in species with different locomotion modes. Since the mechanical environment of limb joints constantly changes with growth (*Hamrick, 1999*), we analyze these histological changes at different ontogenetic stages, from metamorphs to adults in order to record the effects that overuse causes on the tissues. To address these issues, juvenile and adult frog specimens were trained on a treadmill belt on a daily basis for 2 months. Based on previous work we predict (a) that knee tissues of frogs subjected to excessive exercise will deviate from the normal and healthy state, (b) higher morphological damage in adults

than in juveniles, considering that younger tissues are presumably more adaptable (*Clark & Ogden, 1983*; *Bailey, 2001*; *Brack et al., 2007*; *Senan et al., 2011*; *Thampatty & Wang, 2018*) and (c) more alteration in tissues in jumper species compared to walkers ones.

# MATERIALS AND METHODS

## Specimens

A total of 67 specimens of five frog species were analyzed: 10 juveniles and 10 adults of *Leptodactylus mystacinus* Burmeister 1861, seven juveniles and 12 adults of *Rhinella arenarum* Hensel 1867, seven adults of *Melanophryniscus rubriventris* Vellard 1947, six juveniles and five adults of *L. latinasus* Jiménez de la Espada 1875 and four juveniles and six adults of *Phyllomedusa sauvagii* Boulenger 1882. The juveniles were recognized as individuals who have completed the metamorphosis, because they exhibit traits that indicate the completion of the metamorphosis (See *Gosner, 1960*, characters of the mouth and complete tail reabsorption), but they have not reached the adulthood size or sexual maturity indicated by secondary sexual characteristics. Thus, assessment of sexual maturity and identification of adult males were based on the presence of secondary sexual characters (e.g., colored vocal sacs, nuptial excrescences); sexual maturity of females was based on examination of the gonads. Previous experimental studies were performed with laboratory animals (*Vilarta & De Campos Vidal, 1989*; *Ni et al., 2015*; *Nagai et al., 2016*), which allows the use of a high number of specimens. However, laboratory animal's exhibit restricted movements during their lives due to generally being confined to a box. In the present study, we used animals collected in the field, thus allowing us to assume that their histomorphology was determined by their normal conditions of mobility. The disadvantage is that the number of collected specimens is restricted, thus deriving in a small sample size. Specimens were collected during summer in Tucumán (Res. No.13–16), Salta (Res. No. 0308/14) and Jujuy (Res. No. 21/2012) provinces, Argentina. They were housed at Instituto de Herpetología of the Fundación Miguel Lillo, in individual terrariums (30 × 20 × 25 cm), where they moved freely, and under laboratory controlled conditions (temperature 24–29 °C). They were fed ad libitum with living insects (ants, crickets, cockroaches and worms). All the specimens were healthy and without signs of previous injuries. Animals were weighed with a digital scale (±0.01 gr.; Cen-Tech, San Diego, CA, USA) and sized snout-vent length (SVL) with a digital caliper (±0.01 mm.; CD-30C and CD-15B; Mitutoyo, Kanagawa, Japan) before and after the experiments. Since both SVL and weight were similar before and after the trials (±6 mm), we inform only the initial data (Table 1). Specimens of each species with both weight and SVL similar to those used for experiments were selected as control.

## Experimental design

To observe the effect of the mechanical stress provoked by overuse of knee tissues, trials were performed on a treadmill belt, following the current protocols designed to this end (*Kovanen, Suominen & Peltonen, 1987*; *Birch et al., 1999*; *Ni et al., 2015*; *Gao et al., 2017*; *Thampatty & Wang, 2018*). We defined "overuse" as the excessive use of the joint when the frog is "over-stimulated" to move. In nature, they often stay still

**Table 1  Specimens used in the overused experiments.**

| Species | Identification number (MCV) | SVL (mm) | Weight (gr) | Mean ± SD time in 1 day (min) | Total time (min) | Mean ± SD distance in 1 day (m) | Total distance (m) |
|---|---|---|---|---|---|---|---|
| *Leptodactylus latinasus* | 382 | 19.23 (cj) | 1.1 | 0 | 0 | 0 | 0 |
| | 262 | 21.81 (cj) | 1.3 | 0 | 0 | 0 | 0 |
| | 305 | 22.94 (cj) | 1.3 | 0 | 0 | 0 | 0 |
| | 109 | 17.93 (ej) | 0.5 | 6.39 ± 2.45 | 230.93 | 10.47 ± 4.02 | 378.26 |
| | 108 | 20.36 (ej) | 1.3 | 6.02 ± 2.79 | 216 | 9.86 ± 4.5 | 353.58 |
| | 110 | 21.06 (ej) | 1.2 | 6.42 ± 2.02 | 225.16 | 10.52 ± 3.3 | 368.8 |
| | 409 | 26.91 (ca) | 2.1 | 0 | 0 | 0 | 0 |
| | 453 | 30.41 (ca) | 2.1 | 0 | 0 | 0 | 0 |
| | 451 | 30.86 (ca) | 2.6 | 0 | 0 | 0 | 0 |
| | 452 | 27.82 (ea) | 2.2 | 10.10 ± 1.22 | 681.43 | 16.55 ± 2 | 1116.18 |
| | 450 | 29.16 (ea) | 2 | 10.15 ± 0.76 | 789 | 16.63 ± 1.25 | 1291.82 |
| *Leptodactylus mystacinus* | 82 | 18.9 (cj) | 0.6 | 0 | 0 | 0 | 0 |
| | 42 | 16.51 (cj) | 0.7 | 0 | 0 | 0 | 0 |
| | 50 | 15.99 (cj) | 0.6 | 0 | 0 | 0 | 0 |
| | 52 | 18.53 (cj) | 0.5 | 0 | 0 | 0 | 0 |
| | 49 | 17.61 (cj) | 0.5 | 0 | 0 | 0 | 0 |
| | 98 | 19.58 (ej) | 0.6 | 1.82 ± 0.83 | 199 | 2.98 ± 1.36 | 325.38 |
| | 57 | 19.95 (ej) | 0.7 | 1.55 ± 0.45 | 35.23 | 2.55 ± 0.73 | 57.71 |
| | 97 | 19.58 (ej) | 0.7 | 2.58 ± 1.15 | 303 | 4.24 ± 1.89 | 496.28 |
| | 99 | 18.85 (ej) | 0.5 | 1.61 ± 0.74 | 151.4 | 2.65 ± 1.22 | 248.04 |
| | 100 | 17.58 (ej) | 0.6 | 1.72 ± 0.64 | 161.1 | 2.83 ± 1.05 | 263.88 |
| | 537 | 53.82 (ca) | 20 | 0 | 0 | 0 | 0 |
| | 538 | 56.83 (ca) | 20 | 0 | 0 | 0 | 0 |
| | 539 | 57.48 (ca) | 20 | 0 | 0 | 0 | 0 |
| | 540 | 58.57 (ca) | 25 | 0 | 0 | 0 | 0 |
| | 541 | 55.29 (ca) | 20 | 0 | 0 | 0 | 0 |
| | 532 | 58.23 (ea) | 20 | 10 | 1,140 | 16.38 | 1,867.32 |
| | 533 | 57.13 (ea) | 45 | 10 | 1,140 | 16.38 | 1,867.32 |
| | 534 | 57.81 (ea) | 50 | 10 | 1,140 | 16.38 | 1,867.32 |
| | 535 | 57.16 (ea) | 45 | 10 | 1,140 | 16.38 | 1,867.32 |
| | 536 | 56.8 (ea) | 25 | 10 | 1,140 | 16.38 | 1,867.32 |
| *Melanophryniscus rubriventris* | 438 | 37.60 (ca) | 4.6 | 0 | 0 | 0 | 0 |
| | 128 | 37.80 (ca) | 3.9 | 0 | 0 | 0 | 0 |
| | 434 | 34.95 (ca) | 3.9 | 0 | 0 | 0 | 0 |
| | 437 | 35.66 (ca) | 3.6 | 0 | 0 | 0 | 0 |
| | 436 | 33.42 (ea) | 3.5 | 10.04 ± 0.77 | 826.5 | 16.45 ± 1.2 | 1,353.8 |
| | 439 | 36.21(ea) | 3.9 | 8.13 ± 2.3 | 794.4 | 13.32 ± 3.78 | 1,301.28 |
| | 440 | 42 (ea) | 4.9 | 10.15 ± 0.38 | 947.56 | 16.62 ± 0.6 | 1,552.1 |

*(Continued)*

| Species | Identification number (MCV) | SVL (mm) | Weight (gr) | Mean ± SD time in 1 day (min) | Total time (min) | Mean ± SD distance in 1 day (m) | Total distance (m) |
|---|---|---|---|---|---|---|---|
| Phyllomedusa sauvagii | 76 | 25.78 (cj) | 2.7 | 0 | 0 | 0 | 0 |
| | 115 | 27.66 (cj) | 2.5 | 0 | 0 | 0 | 0 |
| | 67 | 25.78 (ej) | 2.1 | 4.91 ± 1.56 | 247 | 8.04 ± 2.56 | 405.03 |
| | 81 | 26.85 (ej) | 1.8 | 5.61 ± 1.08 | 239 | 9.20 ± 1.78 | 391.56 |
| | 78 | 70.79 (ca) | 18.8 | 0 | 0 | 0 | 0 |
| | 88 | 63. 87 (ea) | 14.7 | 9.19 ± 1.51 | 270 | 16.29 ± 1.61 | 442.26 |
| | 107 | 60 (ea) | 16.8 | 9.79 ± 0.70 | 416.46 | 16.04 ± 1.16 | 1,501 |
| | 444 | 69.9 (ea) | 20.1 | 9.84 ± 0.64 | 770.9 | 12.73 ± 1.06 | 1262.8 |
| | 443 | 70 (ea) | 18.9 | 10.14 ± 0.58 | 794 | 16.61 ± 0.96 | 1300.57 |
| Rhinella arenarum | 104 | 52. 75 (cj) | 14.5 | 0 | 0 | 0 | 0 |
| | 455 | 49.43 (cj) | 7.3 | 0 | 0 | 0 | 0 |
| | 456 | 52.56 (cj) | 12.5 | 0 | 0 | 0 | 0 |
| | 446 | 24.85 (ej) | 1.7 | 9.99 ± 1.41 | 903.08 | 16.37 ± 2.3 | 1479.25 |
| | 445 | 26.32 (ej) | 1.5 | 10.07 ± 0.23 | 915.1 | 16.49 ± 0.38 | 1498.9 |
| | 447 | 43.43(ej) | 6 | 10.04 ± 0.56 | 914.7 | 15.79 ± 0.45 | 1497.7 |
| | 105 | 54. 33 (ej) | 14.4 | 6.43 ± 1.53 | 664.6 | 10.53 ± 2.5 | 1088.64 |
| | 526 | 87.62 (ca) | 105 | 0 | 0 | 0 | 0 |
| | 527 | 80.09 (ca) | 60 | 0 | 0 | 0 | 0 |
| | 528 | 87.79 (ca) | 95 | 0 | 0 | 0 | 0 |
| | 529 | 90.69 (ca) | 95 | 0 | 0 | 0 | 0 |
| | 530 | 93.42 (ca) | 100 | 0 | 0 | 0 | 0 |
| | 531 | 83.45 (ca) | 75 | 0 | 0 | 0 | 0 |
| | 520 | 85.91(ea) | 55 | 10 | 1,140 | 16.38 | 1,867.32 |
| | 521 | 91.46 (ea) | 80 | 10 | 1,140 | 16.38 | 1,867.32 |
| | 522 | 109.6 (ea) | 100 | 10 | 1,140 | 16.38 | 1,867.32 |
| | 523 | 82.25 (ea) | 60 | 10 | 1,140 | 16.38 | 1,867.32 |
| | 524 | 101.61 (ea) | 95 | 10 | 1,140 | 16.38 | 1,867.32 |
| | 525 | 80.16 (ea) | 55 | 10 | 1,140 | 16.38 | 1,867.32 |

**Note:**

ej, experimental juvenile; cj, control juvenile; ea, experimental adult; ca, control adult; SVL, snout-vent length; MCV, field number of Miriam Corina Vera.

(Reilly et al., 2015) unless they need to move to escape from predators, find food, defend territories or find couple to mate (Nauwelaerts, Stamhuis & Aerts, 2005). The treadmill belt is one meter long and exhibits a flat surface. It is covered with a transparent polycarbonate box to prevent the escape of animals, while its lateral wall is covered with scaling paper (Fig. 1). The specimens were kept in captivity for 2 days before the performance trials. Specimens were stimulated to jump (jumper species: *L. latinasus* and *L. mystacinus*, Jorgensen & Reilly, 2013; Fabrezi et al., 2014; specie: *R. arenarum*; Abdala et al., 2018) or walk (walker species: *M. rubriventris*, *P. sauvagii*, Manzano et al., 2013; Fabrezi et al., 2014) by the contact with an elastic band crossed on the treadmill belt (Fig. 1), avoiding the stress that human contact could generate. A total of 34 specimens

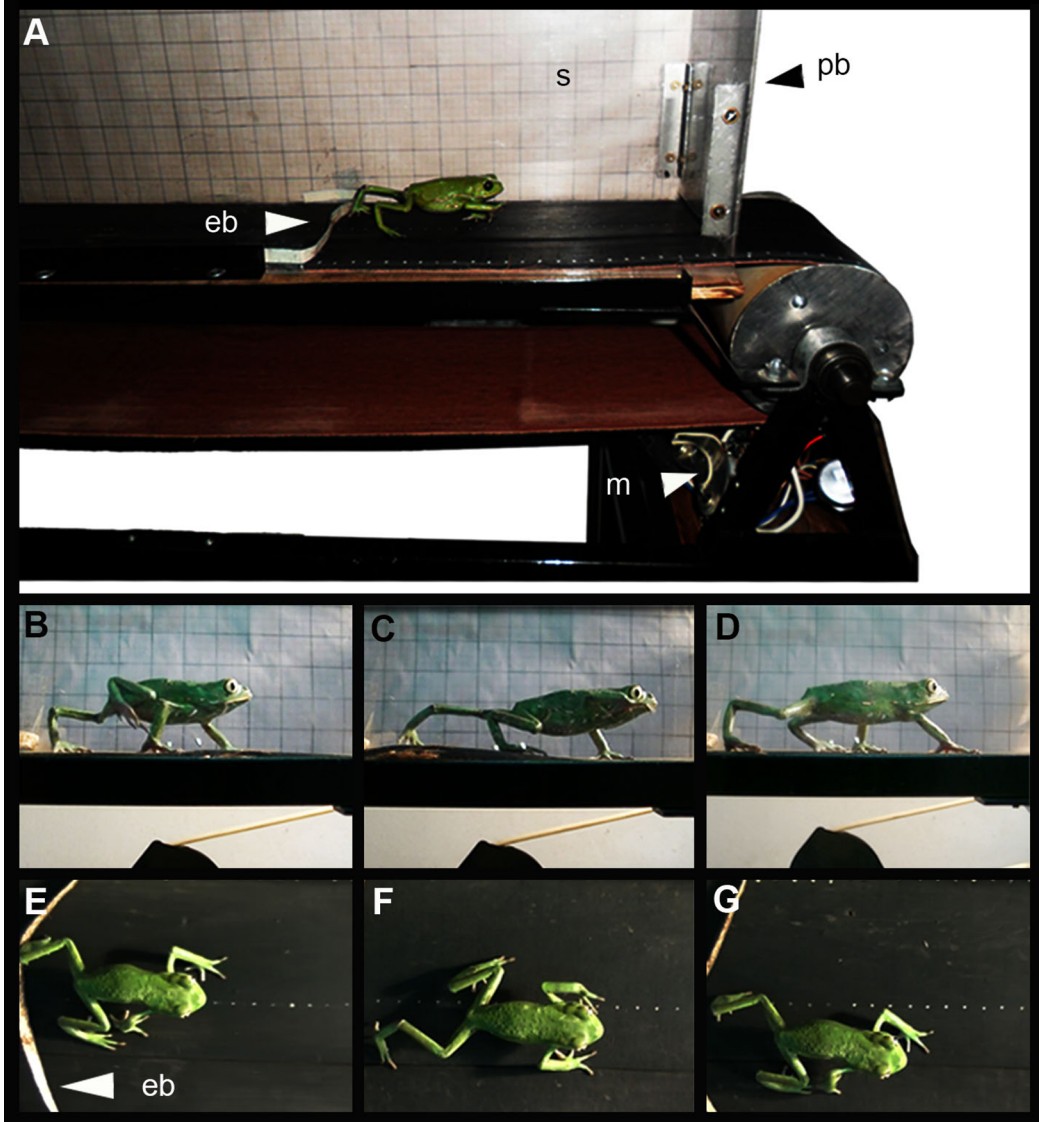

**Figure 1** **(A) Treadmill belt, (B to G) Lateral and dorsal view of a specimen of *Phyllomedusa sauvagii* walking on the treadmill.** Abbreviations: eb, elastic bands; m, motor; s, scale; pb, polycarbonate box. Photo by M.C.V.                                       

were arbitrarily chosen to perform the trials and the remaining specimens were used as control (i.e., did not undergo the exercise routine, Table 1). Trials were performed twice a day, during up to 10 min unless the frog reached the fatigue earlier. Fatigue was defined as the failure to maintain force of a muscle that has been under load, and that is relieved by rest (*Johnson et al., 1996*). During muscular fatigue, a cascade of physiological mechanisms occurs (*Gibson & Edwards, 1985*), provoking muscular pain and dyspnea, which are the principal reasons to stop motion (*Güell, Casan & Giménez, 1996*).

After a series of previous trials, the velocity of the treadmill belt was settled at 2.73 cm/sec (±0.25 cm/sec), being this the maximum value allowing normal movement of specimens. These preliminary trials showed that velocity could be kept constant for all the animals.

Juveniles and adults were exercised at the same velocity allowing to evaluate the effect of the same mechanical stimulus over the joint at different stages. The time and distance of the trials are detailed in Table 1. Experimental and control animals were sacrificed with xylocaine viscous (Lidocaine Hydrochloride 10%), fixed in a 10% formaldehyde solution for 24 h, preserved in alcohol 70%. Selected species for our study are composed by organisms whose life history is subject to *r* selection, often referred to as *r*-strategists. They inhabit temporary ponds, which are unstable and unpredictable environments. They have ability to reproduce quickly, produce many offspring, each of which has a relatively low probability of surviving to adulthood. In the case of *M. rubriventris*, that left not so many offspring, we used fewer specimens. The actual population trends of the species used for this study are LESS CONCERN according to the IUCN red list of threatened species (www.iucnredlist.org, version 2017-3). Apart from softly inducing them to move twice a day, specimens in captivity were maintained clean, healthy, with enough water and food (according to the Amphibian Husbandry Resource Guide: *Poole & Grow (2012)*). In addition, we pay attention to ARRIVE guidelines and to Guidelines for Ethical Conduct in the Care and Use of Nonhuman Animals in Research (CARE). Experiments were approved by the Ethical Committee of Facultad de Medicina, Universidad Nacional de Tucumán (Res. No. 81962-2014).

## Histological analysis

A total of 67 knees, 34 from treated and 33 from control specimens corresponding both to juveniles ($n = 27$) and adults ($n = 40$) (Table 1 and Supplemental Material S1) were extracted and decalcified with a 50% citrate sodium-50% formic acid solution. Samples were immersed in sodium sulfate for 24 h, and then immersed in a mixture of glycerin and acetic acid for 48 h. The material was afterward dehydrated in a graded ethanol series and embedded in Histoplast embedding medium. Seven $\mu$m thick serial sagittal sections were cut with a rotary microtome (Microm HM 325; ThermoFisher Scientific, Waltham, MA, USA) and stained with hematoxiline-eosine and Masson trichrome, the latter allowing to identify collagen fibers. Histological samples were observed under an optical microscope (ICC 50 HD; Leica, Wetzlar, Germany) and photographed with a Nikon Coolpix P6000 digital camera for the diagnosis. The focus was put on the following tissues that integrate the knee-joint: tendons, fibrocartilage, menisci and articular cartilage. Additionally, the growth zone in the diaphysis was examined.

## Tissue alterations score

To assess the effect of the experiments on the connective tissues, five parameters were considered: (i) collagen fibers of the fibrocartilage (ii) collagen fibers of the tendons (iii) roundness of the nuclei of the fibrocartilage (iv) arrangement of the collagen fibers of the menisci and (v) hypertrophic chondrocytes. The structural changes observed were categorized in a scoring system. Histological Scoring is a technique widely used in orthopedic research and clinical veterinary (*Movin et al., 1997*; *O'Driscoll et al., 2001*; *Pritzker et al., 2006*; *Maffulli et al., 2008*; *Pauli et al., 2011*). It is commonly used to show structural qualitative changes of tissue owed to any factor (*Movin et al., 1997*;

**Table 2 Criteria and scores for histological assessment of the connective tissues.**

| Variables | Score |
|---|---|
| **I. *Collagen fiber arrangement of the fibrocartilage*** | |
| The collagen fibers are packaged | 0 |
| The collagen fibers begin to unpack showing a lax configuration | 1 |
| Collagen fiber are more separated showing a very lax configuration | 2 |
| **II. *Roundness of the nuclei of the cells of the fibrocartilage*** | |
| Round nuclei of the fibrocartilage | 0 |
| The nuclei flattened showing a more ovoid shape | 1 |
| The nuclei show a very flat shape | 2 |
| **III. *Collagen fiber arrangement of the tendon*** | |
| The collagen fibers are packaged | 0 |
| The collagen fibers begin to unpack showing a lax configuration | 1 |
| Collagen fiber are more separated showing a very lax configuration | 2 |
| **IV. *Collagen fiber arrangement of the menisci*** | |
| Marked separation of fibers | 0 |
| Collagen fiber becomes more packed | 1 |
| Collagen fibers show a packed arrangement | 2 |
| **V. *Shape of the hypertrophic chondrocytes of the diaphyses*** | |
| The hypertrophic chondrocytes have they typical oval or round shape | 0 |
| The hypertrophic chondrocytes become flattening | 1 |
| The hypertrophic chondrocytes show a very flat shape | 2 |

*Ameye et al., 2002*; *Pritzker et al., 2006*; *Maffulli et al., 2008*; *Pauli et al., 2011*). In this study, two levels of structural or morphological changes were identified, namely, Score 1 for slight changes and Score 2 for severe changes. Score 0 was assigned to those tissues that showed a normal morphology. The tissue scoring is described in Table 2. This grading system is arbitrary and it does not represent fixed stages, however, it is a simple and direct, way to represent the effect of the experiments (*Pritzker et al., 2006*). Since it is as accurately as possible, is one of the most used systems in this context (see quotations above). Tissues were considered normal following *Carvalho (1995)* and *Franchi et al. (2007)* for tendons; *Benjamin et al. (1991)*, *Benjamin, Qin & Ralphs (1995)*, *Benjamin & Ralphs (1998)* and *Carvalho & Felisbino (1999)* for fibrocartilages; *Pauli et al. (2011)* and *Senan et al. (2011)* for menisci and *Pacifici et al. (1990)* for hypertrophic chondrocytes. For an overview of the alteration state of each specimen, scores of each trait were summed up. Specimens with Score 10 were those whose connective tissues exhibited the highest levels of abnormality.

The association of the score of each trait with the treatment, the locomotor mode, the species identity and the stage of the individual were assessed by using multinomial ordinal logistic regression. The multinomial ordinal logistic regression is used for describing and testing hypotheses about associations between an ordered categorical variable (i.e., the alteration tissues) and one or more categorical or continuous predictor variables (i.e., treatment, locomotor mode, stages and species) to predict the probability of occurrence of each category (*Peng, Lee & Ingersoll, 2002*). In the case of the shape of the hypertrophic chondrocyte where only two classes were observed (0 and 2) we used a
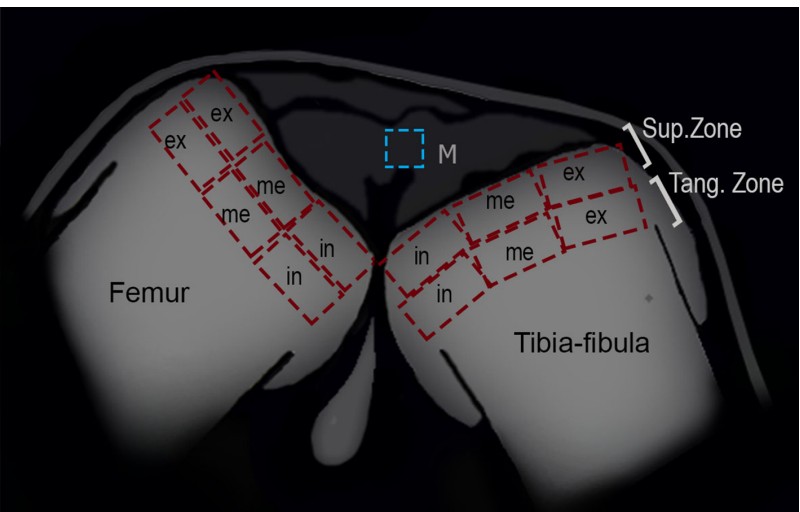

**Figure 2 Schematic representation of the frog knee joint.** Abbreviations: Sup.Zone, superficial zone; Tang.Zone, tangential zone; M, menisci; in, internal zone; me, medial zone; ex, external zone.

simple logistic regression that is adequate to model binomial responses. In all the cases we proposed a set of biologically sensitive models (e.g., lineal combinations of predictors) and we compared to what extent they were supported by our data. The set of competing models were increasingly complex; they included an intercept model, which assumes that the probability of observing any category is identical for all the individuals, a model that only considered the treatment, different lineal combinations of the treatment with the stage, the species and the locomotor mode and the interaction between treatment and the other predictors. Locomotor mode and species were not included in the same model due to their nestedness. The models were compared using their Akaike information criterion (AIC; *Symonds & Moussalli, 2011*). The AIC simultaneously evaluates the level of adjustment of a model (maximum likelihood) with the number of parameters (i.e., an indicator of the complexity of the model). The AIC is not informative by itself and only has utility for compare different models. ∆AIC (delta of AIC) is used as a measurement of the distance of all the models that explain a variable with respect to the model with the best AIC. Due to the nature of our data, we used an AIC corrected by small samples (*Burnham & Anderson, 2002*). The best logistic models were identified by minimum AIC (*Burnham, Anderson & Huyvaert, 2011*). All the statistics analyses were performed with the R studio software (version 0.99.903, 2016).

## Menisci cell quantification

To analyze the density of cells in the menisci, an area of 100 $\mu m^2$ of the knees of 58 specimens (30 treated and 28 control specimens) was selected (Fig. 2). The number of cells was quantified with ImageJ software and the density was calculated ($\rho$ = no of cells/area). The normality and homoscedasticity of the data were tested with Shapiro–Wilk and Levene's tests. Differences in the density of cells between control and treated groups were calculated with a Mann–Whitney $U$-test using R studio software (version 0.99.903, 2016).

## Articular cartilage

To analyze the density of cells in the superficial and tangential zones of the articular cartilage, the knees of 25 (13 treated and 12 control) juvenile specimens were used. Adult epiphyses were not analyzed since their articular cartilage was not visible in our samples. The zones of the epiphyses of the femur and tibia-fibula were divided in three areas: internal, medial and external (Fig. 2). The number of chondrocytes was counted in each region with ImageJ software, and the density was derived ($\rho$ = no of chondrocytes/area). The normality and homoscedasticity of the data were tested with Shapiro–Wilk and Levene's tests. Differences in the density of cells between control and treated groups were calculated with a Mann–Whitney $U$-test. A posteriori Kruskal–Wallis test was made to assess for differences among zones and areas. R studio software (version 0.99.903, 2016) was used for the statistical analyses.

## RESULTS

### Microanatomy of the knee-joint

The normal knee-joint of an anuran juvenile specimen (SVL 22.94 mm) is formed by the joint capsule, the menisci, the epiphyses of the femur and the tibia-fibula, muscles, ligaments and tendons (Fig. 3A). The joint capsule consists of fibrous and dense connective tissue, that is, fibrocartilage and tendons. The tendons exhibit parallel collagen fibers with abundant and round nuclei (Fig. 3B). A spindle-shape fibrocartilage is present in the external surface of the knee over the tibia-fibula epiphysis (Fig. 3A). The cells of the fibrocartilage present spherical nuclei, usually arranged in rows, and collagen fibers arranged in parallel (Fig. 3B). The meniscus is present over and between the two epiphyses; the zones of attachment with the epiphyses (enthesis) are usually fibrocartilaginous (Fig. 3C). The meniscus is fibrocartilaginous, with collagen fibers usually packed or showing a more disordered pattern. The nuclei of the collagen fibers of the meniscus are dispersed or arranged in rows (Fig. 3C). The epiphyses of the femur and the tibia-fibula are cartilaginous and covered by the articular (hyaline) cartilage. The chondrocytes of the articular cartilage are isolated or disposed in isogenous groups. The irrigated osteochondral ligaments are located between the lateral articular cartilage and the periosteal bone of the diaphysis (Fig. 3D). Internally, a cartilaginous graciella sesamoid is found between the femur and the tibia-fibula (Fig. 3E). The m. gracilis major (Fig. 3E), m. extensor cruris brevis, m. vastus internus and m. gastrocnemius are mature at this stage.

The normal knee of an adult specimen of *L. latinasus* (SVL 30.86 mm; Fig. 4) presents a joint capsule with a big fibrocartilage over the surfaces of the tibia-fibula and of the femur (Fig. 4A). The cells of the adult fibrocartilage show round nuclei in rows, and parallel collagen fibers (Fig. 4B), more ordered than in juveniles. Tendons are mature tissues, as evidenced by the parallels fibers and flat nuclei. The menisci are thicker than in juveniles and show areas with packed and laxer collagen fibers (Fig. 4C). The graciella sesamoid is cartilaginous with a center of endochondral ossification. The m. gracilis major is mature (Fig. 4D). The epiphyses are ossified, evidenced by the wide medular cavity with endochondral trabeculae, osteocytes, osteoclasts and blood vessels (Fig. 4E).

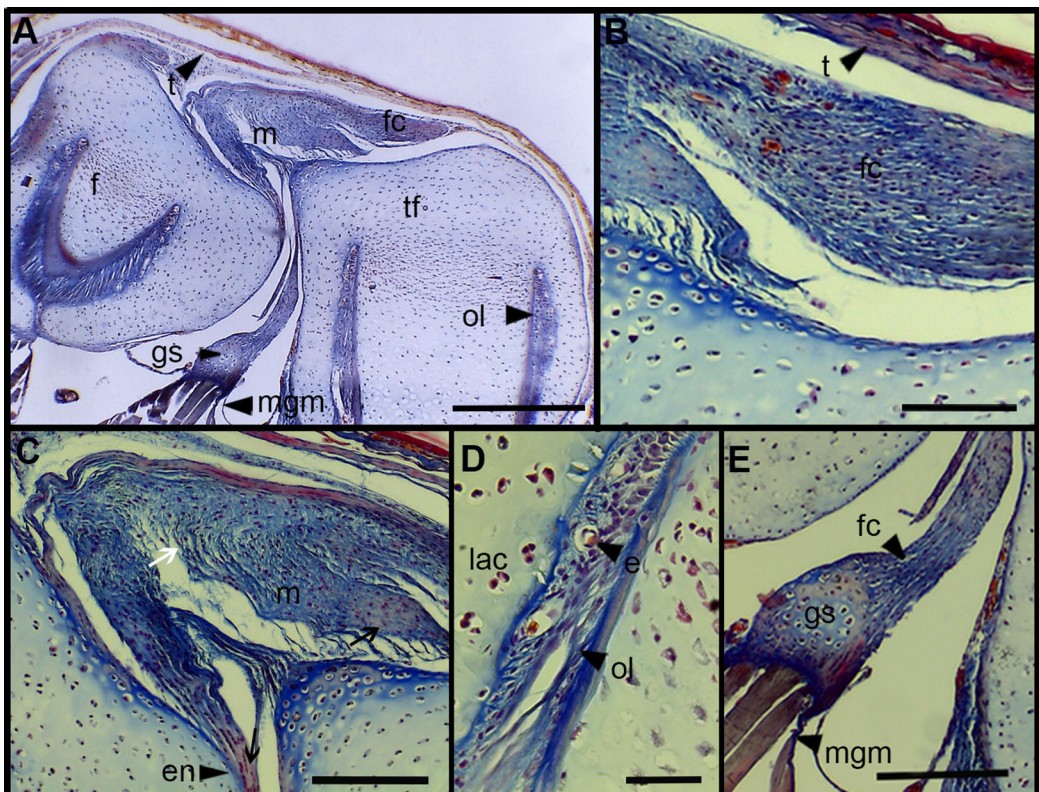

**Figure 3 Histology of the knee-joint of *Leptodactylus latinasus*.** (A) Knee-joint of a juvenile specimen (SVL 22.94 mm), scale bar 500 μm; (B) detail of the fibrocartilage, scale bar 100 μm; (C) detail of the menisci, scale bar 200 μm; (D) detail of the osteochondral ligament, scale bar 50 μm; (E) detail of the graciella sesamoid, scale bar 200 μm. Abbreviations: DP, distal patella; en, enthesis; e, erythrocytes; f, femur; fc, fibrocartilage; gs, graciella sesamoid; lac, lateral articular cartilage; m, menisci; mgm, muscle gracilis major; ol, osteochondral ligament; tf, tibia-fibula; t, tendon. In the menisci, black arrows indicate disperse nuclei, and white arrows indicate nuclei in rows. In the enthesis, the black arrow indicates the fibrocartilaginous tissue.

## Tissue Alterations Scores

Scores are presented in Fig. 5 and a global overview of the altered tissues in juveniles and adults are showed in Fig. 6. The first column in Fig. 5 corresponds to the control specimens and the two right columns correspond to the overuse trials scores.

Normal fibrocartilages were composed by parallel packed collagen fibers between cells (Score 0, Fig. 5A). In treated specimens, there was a gradual separation of the collagen fibers, from a loose (Score 1, Fig. 5B) to a laxer arrangement (Score 2, Fig. 5C). Normal fibrocartilage cells showed round nuclei (Score 0, Fig. 5D). These exhibited a change to oval (Score 1, Fig. 5E) or flattening shape (Score 2, Fig. 5F). Tendons were formed by tightly packed collagen fibers (Score 0, Fig. 5G). Such as in fibrocartilages, tendons showed a gradual disarrangement of their collagen fibers from a loose (Score 1, Fig. 5H) to a very loose pattern (Score 2, Fig. 5I). Normal menisci showed a loose pattern with collagen fibers separated from each other (Score 0, Fig. 5J). The menisci of treated specimens showed a slight (Score 1, Fig. 5K) or a maximum packing (Score 2, Fig. 5E) of their collagen fibers. The hypertrophic chondrocytes of the growth area diaphyses of both femur and
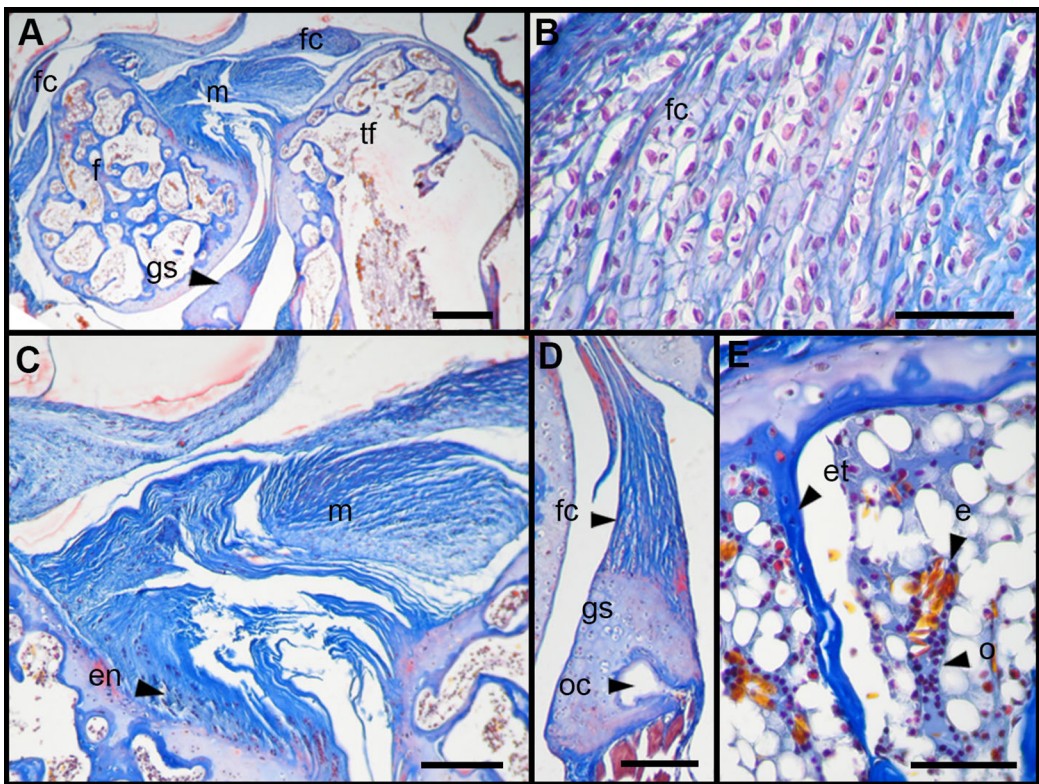

**Figure 4 Histology of the knee-joint of *Leptodactylus latinasus*.** (A) Knee-joint of an adult specimen (SVL 30.86 mm), scale bar 200 μm; (B) detail of the fibrocartilage, scale bar 50 μm; (C) detail of the menisci, scale bar 100 μm; (D) detail of the graciella sesamoid, scale bar 100 μm; (E) detail of the femur epiphyses, scale bar 50 μm. Abbreviations: DP, distal patella; en, enthesis; e, erythrocytes; et, endochondral trabeculas; f, femur; fc, fibrocartilage; gs, graciella sesamoid; m, menisci; o, osteocytes; oc, ossification center; PP, proximal patella; tf, tibia-fibula.

tibia-fibula, normally showed an oval or round shape (Score 0, Fig. 5M). The cells exhibited a drastic change of shape, adopting a flattened shape (Score 2, Fig. 5O). This last feature was observed only in juvenile specimens because in adults the diaphyses are already ossified.

In the joint capsule of experimental juveniles, the cells of the fibrocartilages were very affected (Fig. 7B), and showed a flat or oval nuclei (Table 3). The fibers of the tendon showed a very lax arrangement (Score 2, Fig. 7C) or a slight disarrangement (Score 1, Fig. 7C; Table 3). The menisci fibers presented a packed arrangement (Score 2, Fig. 7D). The hypertrophic chondrocytes of the growth area of the diaphyses showed a severe flattening (Score 2, Fig. 7E; Table 3).

Among adults, the fibrocartilaginous tissue showed more extreme changes (Fig. 7; Table 3). There was a noticeable separation of the collagen fibers of the fibrocartilage (Score 2, Fig. 7F) and the nuclei of the fibrocartilage cells were elongated (Score 2, Fig. 7G; Table 3). Tendons also showed a slight abnormal arrangement with a loose pattern of the collagen fibril (Score 1, Fig. 7H).

The highest total sums of scores were 5 and 7 for one specimen of *L. mystacinus* and one adult specimen of *L. latinasus*, respectively. Both species have a jumper locomotion

| Parameters \ Scores | Control | Overuse trials | |
| --- | --- | --- | --- |
| | 0 | 1 | 2 |
| Arrangement of the collagen fiber of the fibrocartilage | Packed | Loose | Very loose |
| |  |  |  |
| Roundness of the nuclei of the the fibrocartilage cells | Rounded | Oval | Flattened |
| |  |  |  |
| Arrangement of the collagen fiber of the tendon | Packed | Loose | Very loose |
| |  |  |  |
| Arrangement of the collagen fiber of the menisci | Loose | Slightly packed | Packed |
| |  |  |  |
| Shape of the hypertrophic chondrocytes | Oval | Flattened | Very flattened |
| |  |  |  |

**Figure 5  Scoring system showing graduals changes.** Score 0 correspond to control, while Scores 1 and 2 correspond to overused trials. Scale bar 50 μm. (A) Sample of *L. latinasus* (juvenile, SVL 22.94 mm), (B) sample of *Phyllomedusa sauvagii* (adult, SVL 70.79 mm), (C) sample of *L. latinasus* (adult, SVL 29.16 mm), (D) sample of *L. latinasus* (adult, SVL 30.86 mm), (E) sample of *L. latinasus* (juvenile, SVL 22.94 mm), (F) sample of *L. latinasus* (adult, SVL 29.16 mm), (G) sample of *L. latinasus* (adult, SVL 30.41 mm), (H) sample of *Rhinella arenarum* (juvenile, SVL 52.75 mm), (I) sample of *R. arenarum* (juvenile, SVL 54.33 mm), (J) sample of *Leptodactylus mystacinus* (juvenile, SVL 17.58 mm), (K) sample of *L. mystacinus* (juvenile, SVL 19.58 mm), (L) sample of *L. latinasus* (juvenile, SVL 22.94 mm), (M) sample of *L. mystacinus* (juvenile, SVL 16.51 mm), (N) black bar: non observed, (O) sample of *L. mystacinus* (juvenile, SVL 19.58 mm).     

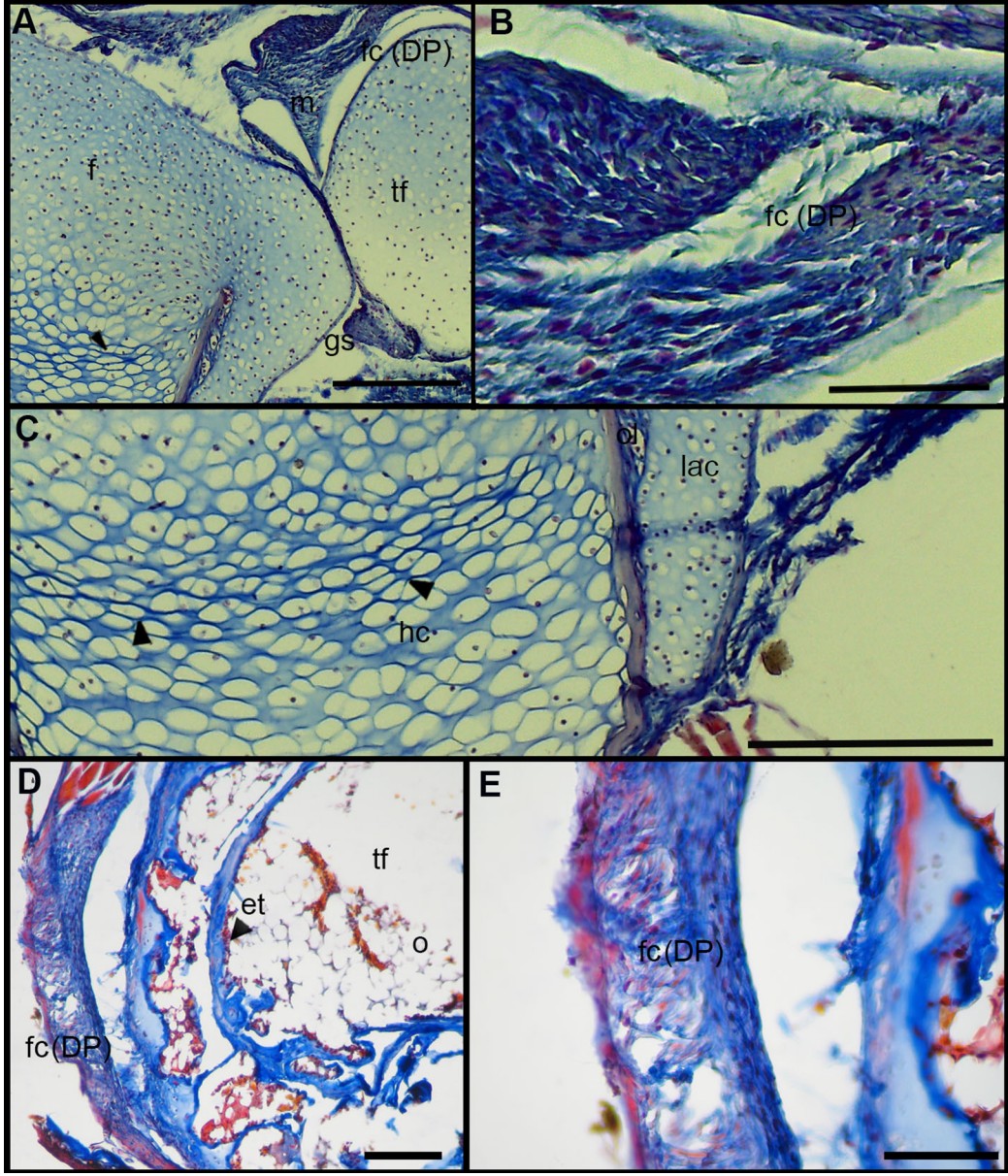

**Figure 6 Histology of the knee-joint of treatment specimens.** (A) Knee-joint of juvenile of *L. mystacinus*, scale bar 200 μm, (B) detail of the fibrocartilage of *L. mystacinus*, showing the disarrangement of the collagen fibers and oval and flatted nucleus, scale bar 50 μm, (C) detail of the femur diaphysis of *L. mystacinus*, showing flatted hypertrophic chondrocytes, scale bar 200 μm, (D) tibia-fibula of an adult of *L. latinasus*, scale bar 100 μm, (E) detail of the an altered fibrocartilage of an adult of *L. latinasus*, scale bar 50 μm. Abbreviations: DP, distal patella; et, endochondral trabeculae; f, femur; fc, fibrocartilage; gs, graciella sesamoid; hc, hypertrophic chondrocyte; lac, lateral articular cartilage; m, menisci; o, osteocytes; ol, osteochondral ligament; tf, tibia-fibula.

mode. The walker species also show tissue alterations, but the sum of their scores was 4 in a juvenile and an adult specimen of *P. sauvagii* and 4 in adults of *M. rubriventris*. A juvenile of *R. arenarum* have a total score of 4 and an adult a total score of 6. Scores distribution among species is detailed in Table 3 and Fig. 7.

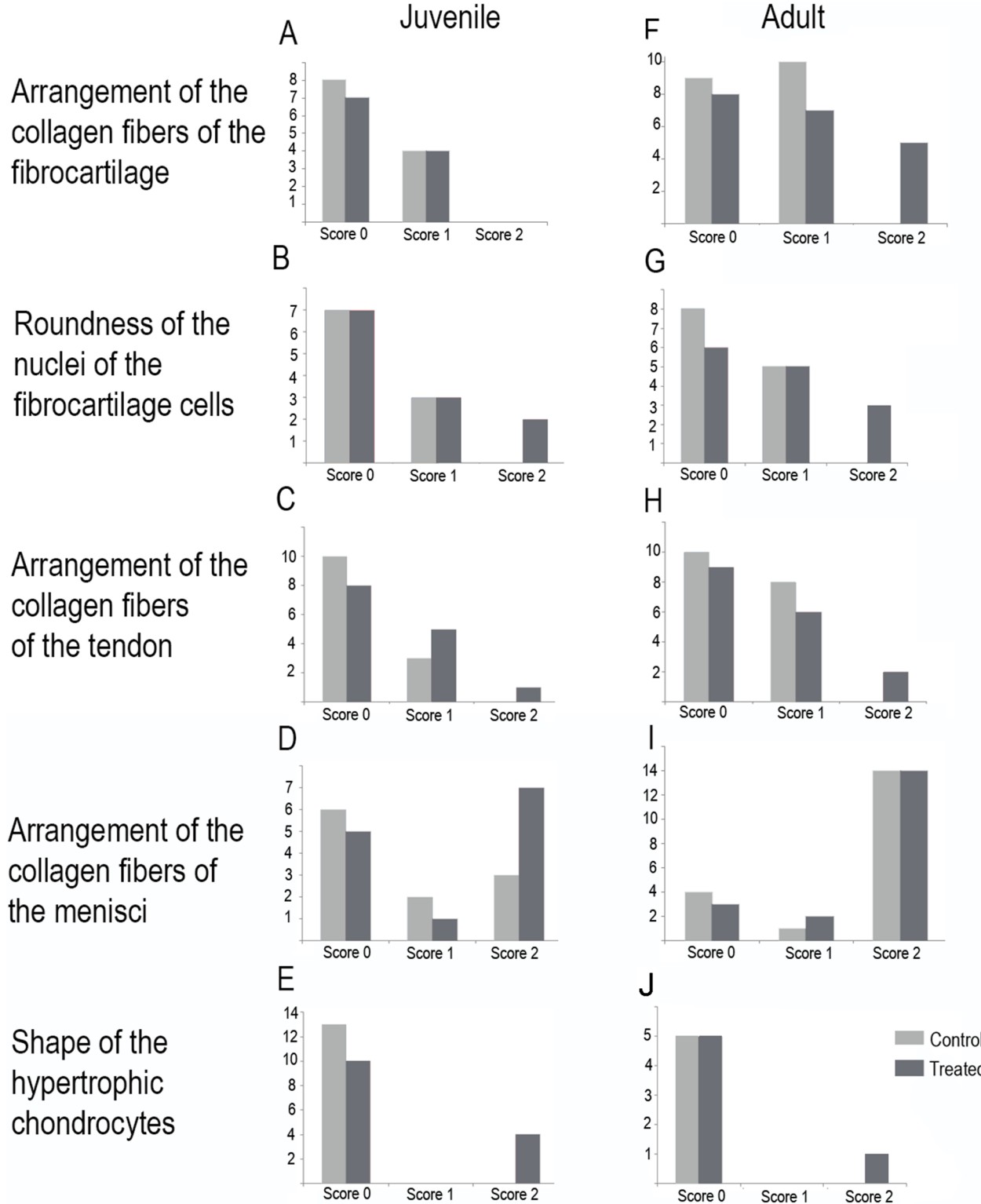

**Figure 7 Distribution of the scores in juveniles (A–E) and adults (F–J).** (A and F) arrangement of the collagen fibers of the fibrocartilage, (B and G) roundness of the nuclei of the fibrocartilage cells, (C and H) arrangement of the collagen fibers of the tendons, (D and I) arrangement of the collagen fibers of the menisci, (E and J) shape of the hypertrophic chondrocytes. Light gray bars correspond to control specimens and dark gray bar correspond to treated specimens. *X*-axis: scores, *Y*-axis: number of specimens.

**Table 3  Distribution of the scores in treated (hypermobilized) and control juveniles and adults.**

| Juveniles | Controls (n = 13) | | | Treated (n = 14) | | |
|---|---|---|---|---|---|---|
| Scores | 0 | 1 | 2 | 0 | 1 | 2 |
| Collagen fiber arrangement of the fibrocartilage* | 8 (L.m, L.l, R.a) | 4 (L.m, P.s) | 0 | 7 (L.m, L.l, R.a) | 4 (L.m, L.l, R.a) | 0 |
| Roundness of the nuclei of the cells of the fibrocartilage* | 7 (L.m, L.l, P.s, R.a) | 3 (L.l, L.m) | 0 | 7 (L.m, L.l, P.s, R.a) | 3 (L.m, L.l) | 2 (P.s) |
| Collagen fiber arrangement of the tendon | 10 (L.m, L.l, P.s) | 3 (R.a) | 0 | 8 (L.m, L.l, P.s) | 5 (L.m, L.l, R.a) | 1 (R.a) |
| Collagen fiber arrangement of the menisci* | 6 (L.l, L.m, R.a, P.s) | 2 (L.m, L.l) | 3 (L.m, L.l, P.s) | 5 (L.m, L.l, P.s) | 1 (L.m) | 7 (L.m, L.l, R.a, P.s) |
| Shape of the hypertrophic chondrocytes of the diaphyses | 13 (L.m, L.l, P.s, R.a) | 0 | 0 | 10 (L.m, L.l, P.s, R.a) | 0 | 4 (L.m) |

| Adults | Controls (n = 20) | | | Treated (n = 20) | | |
|---|---|---|---|---|---|---|
| Scores | 0 | 1 | 2 | 0 | 1 | 2 |
| Collagen fiber arrangement of the fibrocartilage* | 9 (L.l, L.m, P.s, R.a, M.r) | 10 (L.l, L.m, P.s, R.a, M.r) | 0 | 8 (L.m, P.s, R.a, M.r) | 7 (L.m, P.s, M.r, R.a) | 5 (L.l, R.a) |
| Roundness of the nuclei of the cells of the fibrocartilage* | 8 (L.l, L.m, P.s, M.r) | 5 (L.l, M.r) | 0 | 6 (L.m, P.s) | 5 (L.l, P.s, M.r) | 3 (L.l, L.m, P.s) |
| Collagen fiber arrangement of the tendon* | 10 (L.l, L.m, R.a, M.r) | 8 (L.m, R.a, M.r) | 0 | 9 (L.m, P.s, M.r) | 6 (L.l, L.m, R.a, M.r) | 2 (R.a) |
| Collagen fiber arrangement of the menisci* | 4 (L.l, L.m) | 1 (L.m) | 14 (L.l, L.m, P.s, R.a, M.r) | 3 (L.m, R.a, P.s) | 2 (L.m, R.a) | 14 (L.l, L.m, P.s, R.a, M.r) |
| Shape of the hypertrophic chondrocytes of the diaphysis* | 5 (R.a) | 0 | 0 | 5 (R.a) | 0 | 1 (P.s) |

**Notes:**
(0) Normal; (1) Slightly abnormal; (2) Abnormal. L.m, *Leptodactylus mystacinus*; L.l, *Leptodactylus latinasus*; M.r, *Melanophryniscus rubriventris*; R.a, *Rhinella arenarum*; P.s, *Phyllomedusa sauvagii*.
* Features that were not observable in all the specimens.

Table 4 Resume of multinomial and binomial logistic models (for hypertrophic chondrocytes) selections under AICc (Akaike's Information Criterion, adjusted for small sample size) for the five parameters analyzed.

| Models | mFFm.st.tr | mFFm.sp.tr | mFFm | mFFm.md.st.tr | mFFm.int2 | mFFm.md.tr | mFFm.sp.st.tr | mFFm.int3 | mFFm.int1 | mFFm.sp.tr |
|---|---|---|---|---|---|---|---|---|---|---|
| dAICc | 0.0 | 2.0 | 2.0 | 2.2 | 2.3 | 4.3 | 4.9 | 5.2 | 6.0 | 6.0 |
| df | 4 | 3 | 2 | 5 | 5 | 5 | 8 | 5 | 11 | 7 |
| Models | mNFm.sp.tr | mNFm.sp.st.tr | mNFm.md.tr | mNFm.int3 | mNFm.md.st.tr | mNFm.tr | mNFm.int1 | mNFm | mNFm.st.tr | mNFm.int2 |
| dAICc | 0.0 | 2.1 | 4.7 | 6.1 | 6.7 | 9.4 | 9.7 | 9.9 | 11.6 | 13.6 |
| df | 7 | 8 | 4 | 5 | 5 | 3 | 11 | 2 | 4 | 5 |
| Models | mFTm.sp.tr | mFTm.sp.st.tr | mFTm.int1 | mFTm.md.st.tr | mFTm.md.tr | mFTm.int3 | mFTm | mFTm.tr | mFTm.st.tr | mFTm.int2 |
| dAICc | 0.0 | 0.4 | 6.8 | 37.8 | 39 | 40.9 | 45.2 | 46.1 | 47.5 | 49.5 |
| df | 7 | 8 | 11 | 5 | 4 | 5 | 2 | 3 | 4 | 5 |
| Models | mFMm.sp.st.tr | mFMm.sp.tr | mFMm.md.st.tr | mFMm.int3 | mFMm.st.tr | mFMm.int2 | mFMm.md.tr | mFMm.int1 | mFMm | mFMm.tr |
| dAICc | 0.0 | 2.3 | 2.5 | 3.5 | 3.6 | 5.9 | 6.0 | 6.4 | 6.9 | 9.1 |
| df | 8 | 7 | 5 | 5 | 4 | 5 | 4 | 11 | 2 | 3 |
| Models | mCH.sp.st.tr | mCH.sp.tr | mCH.tr | mCH.st.tr | mCH.int1 | mCH.int4 | mCH | | | |
| dAICc | 0.0 | 1.0 | 7.1 | 9.2 | 10.1 | 11.5 | 12.0 | | | |
| df | 6 | 5 | 2 | 3 | 8 | 7 | 1 | | | |

Note:
In bold the model selected. mFFm, fibrocartilage fibers; mNFm, nuclei of the fibrocartilage; mFTM, tendon fibers; mFMm, menisci fibers; mCHm, hypertrophic chondrocytes; st, stage; sp, species; tr, treatment; md, locomotor mode; int1, intercept sp*tr; int2, intercept st*tr; int3, intercept md*tr; int4, intercept st*sp.

In all the analyses the models that included the treatment outperformed the intercept model (more than two units of difference between their AIC). Some of the better models included also the stage (in fibrocartilage collagen fibers), the species (in fibrocartilage nuclei and collagen fiber of the tendon), and the stage and species (in menisci fibers and hypertrophic chondrocytes) (Table 4 and Supplemental Material S2). Neither the locomotor mode nor the interactions between explanatory variables were included in the winning models in any of the analyses. The probabilities estimated through the best models are detailed in Table 5. The probability of finding an alteration in the arrangement of the fibrocartilage fibers (Score 1 or 2) increase after the treatment in juveniles and adults. The latter stage doubles the probability of presenting the Score 2, from 0.07 to 0.14 (Table 5). The probability of showing a shape change in the nuclei of this tissue varies among the species and the treatment. The probability of showing the highest alteration (Score 2) triples in *L. latinasus*, *P. sauvagii* and *M. rubriventris*. No differences in the probability were found between the stages. Regarding tendons, the probability of showing alteration after the treatment also varies between species, in *L. latinasus* and *L. mystacinus* the probabilities of showing the Score 1 doubles, and in *R. arenarum* the probability of showing the Score 2, triples. For menisci fibers arrangement there is a slight effect of the

**Table 5 Estimates of probabilities from the best models of multinomial and binomial logistic regression of: do not suffer a tissue alteration (score 0), suffer a slight (score 1) or a high (score 2) issue alteration.**

| | Stage | Treatment | Species | Prob. score 0 | Prob. score 1 | Prob. score 2 |
|---|---|---|---|---|---|---|
| Collagen fiber arrangement of the fibrocartilage | Juvenile | Control | | 0.76 | 0.20 | 0.02 |
| | | Treated | | 0.59 | 0.35 | 0.05 |
| | Adult | Control | | 0.52 | 0.40 | 0.07 |
| | | Treated | | 0.32 | 0.52 | 0.14 |
| Roundness of the nuclei of the cells of the fibrocartilage | | Control | L. latinasus | 0.44 | 0.48 | 7.63e-02 |
| | | | L. mystacinus | 0.92 | 6.89e-02 | 5.23e-03 |
| | | | R. arenarum | 1.00 | 7.95e-09 | 5.604e-10 |
| | | | P. sauvagii | 0.36 | 0.53 | 0.10 |
| | | | M. rubriventris | 0.41 | 0.50 | 8.58e-02 |
| | | Treated | L. latinasus | 0.16 | 0.58 | 0.25 |
| | | | L. mystacinus | 0.75 | 0.22 | 2.09e-02 |
| | | | R. arenarum | 1.00 | 3.23e-08 | 2.27e-09 |
| | | | P. sauvagii | 0.12 | 0.56 | 0.31 |
| | | | M. rubriventris | 0.14 | 0.57 | 0.27 |
| Collagen fiber arrangement of the tendon | | Control | L. latinasus | 0.89 | 0.10 | 1.58e-10 |
| | | | L. mystacinus | 0.95 | 4.77e-02 | 6.95e-11 |
| | | | R. arenarum | 2.41e-08 | 0.94 | 5.43e-02 |
| | | | P. sauvagii | 1.0 | 7.39e-10 | 0.00 |
| | | Treated | L. latinasus | 0.52 | 0.47 | 1.26e-09 |
| | | | L. mystacinus | 0.71 | 0.28 | 5.54e-10 |
| | | | R. arenarum | 3.03e-09 | 0.68 | 0.31 |
| | | | P. sauvagii | 1.00 | 5.88e-09 | 0.00 |
| Collagen fiber arrangement of the menisci | Juvenile | Control | L. latinasus | 0.29 | 0.21 | 0.48 |
| | | | L. mystacinus | 0.64 | 0.17 | 0.18 |
| | | | R. arenarum | 0.25 | 0.20 | 0.54 |
| | | | P. sauvagii | 0.25 | 0.20 | 0.54 |
| | | Treated | L. latinasus | 0.29 | 0.21 | 0.49 |
| | | | L. mystacinus | 0.64 | 0.17 | 0.18 |
| | | | R. arenarum | 0.25 | 0.20 | 0.54 |
| | | | P. sauvagii | 0.25 | 0.20 | 0.54 |
| | Adult | Control | L. latinasus | 0.11 | 0.12 | 0.75 |
| | | | L. mystacinus | 0.35 | 0.21 | 0.42 |
| | | | R. arenarum | 9.52e-02 | 0.10 | 0.79 |
| | | | P. sauvagii | 9.52e-02 | 0.10 | 0.79 |
| | | | M. rubriventris | 1.20e-08 | 1.75e-08 | 1.00 |
| | | Treated | L. latinasus | 0.11 | 0.12 | 0.76 |
| | | | L. mystacinus | 0.35 | 0.21 | 0.42 |
| | | | R. arenarum | 9.42e-02 | 0.10 | 0.79 |
| | | | P. sauvagii | 9.43e-02 | 0.10 | 0.79 |
| | | | M. rubriventris | 1.19e-08 | 1.73e-08 | 1.00 |

(Continued)

| | Stage | Treatment | Species | Prob. score 0 | Prob. score 1 | Prob. score 2 |
|---|---|---|---|---|---|---|
| Shape of the hypertrophic chondrocytes | Juvenile | Control | L. latinasus | 1.00 | – | 1.93e-20 |
| | | | L. mystacinus | 1.00 | – | 3.51e-10 |
| | | | R. arenarum | 1.00 | – | 1.76e-38 |
| | | | P. sauvagii | 1.00 | – | 5.59e-10 |
| | | Treated | L. latinasus | 1.00 | – | 8.23e-39 |
| | | | L. mystacinus | 0.2 | – | 0.80 |
| | | | R. arenarum | 1.00 | – | 2.02e-28 |
| | | | P. sauvagii | 1.00 | – | 6.46e-10 |
| | Adult | Control | L. latinasus | 0.98 | – | 0.02 |
| | | | L. mystacinus | 0.00 | – | 1.00 |
| | | | R. arenarum | 1.00 | – | 3.00e-20 |
| | | | P. sauvagii | 0.91 | – | 0.08 |
| | | Treated | L. latinasus | 0.00 | – | 1.00 |
| | | | L. mystacinus | 0.00 | – | 1.00 |
| | | | R. arenarum | 1.00 | – | 3.45e-10 |
| | | | P. sauvagii | 0.00 | – | 1.00 |

**Note:**
The best models include: the treatment and the stage (in fibrocartilage collagen fibers), the treatment and the species (in fibrocartilage nuclei and collagen fiber of the tendon), and the treatment, stage and species (in menisci fibers and hypertrophic chondrocytes).

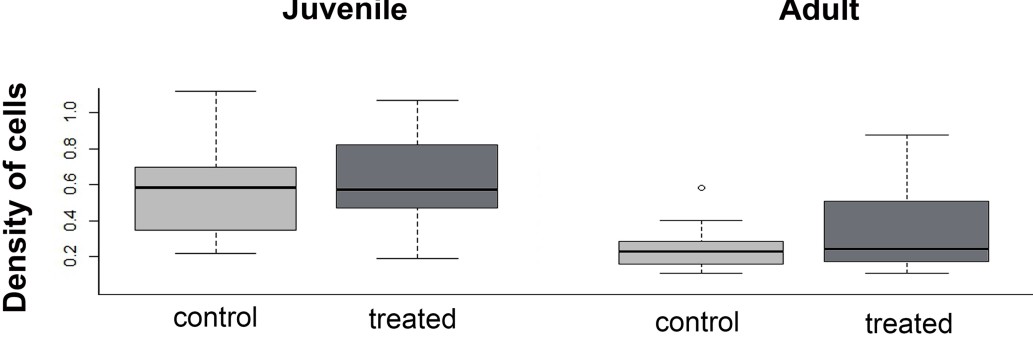

**Figure 8 Boxplot of the meniscus cell density in juvenile and adults, in control and treated groups.**

treatment and a strong effect of the stage (e.g., the probability of observing Score 2 in *L. mystacinus* doubles between stages, from 0.18 to 0.42). For hypertrophic chondrocytes shape the probability of present alteration after the treatment varies also between stages and species. It is higher in juveniles of *L. mystacinus* and adults of *L. latinasus* and *P. sauvagii* (Table 5).

## Menisci cell quantification

The density of cells in the menisci was similar between experimental (i.e., over-exercised) and control individuals, both in juveniles ($U = 136$; $p = 0.43$) and adults ($U = 280.5$; $p = 0.98$, Fig. 8). Both in control and experimental specimens, the density of cells in the menisci was lower in adults than in juveniles.

### Articular cartilage cell quantification

In the articular cartilage of juvenile stages, there were no significant differences in the density of chondrocytes between control and experimental (over exercised) individuals, neither in the femur ($U = 2.578$; $p = 0.38$) nor in the tibia-fibula ($U = 3.269$, $p = 0.08$) (Table 3).

## DISCUSSION

Our results partially sustained the proposed predictions: tissues showed certain degree of deviation from the healthy state. The collagen fibers of the tendon and fibrocartilage were the knee-joint tissues showing phenotypical changes after the overuse trials. The fibers of the menisci and the fibrocartilage showed the highest structural alterations in adults. It should be noted that although evident, all these alterations are gradual. Finally, all locomotor modes showed similar response to the trials, contradicting thus our third prediction.

We infer that adults are more vulnerable to suffer morphological changes after experimental trials. Interestingly, some jumper specimens present the highest score. We observed a severe disarrangement of the collagen fibers of the fibrocartilage located over the tibia-fibula among experimental adults, while most experimental juveniles showed only a slight disarrangement. Interestingly, tadpoles with reduced mobility also presented a high disarrangement of the collagen fibers of the fibrocartilage (*Abdala & Ponssa, 2012*). The distribution and orientation of the collagen fibers are well-adapted to their mechanical role (*Aspden, Yarker & Hukins, 1985*; *Ghosh & Taylor, 1987*; *Vilarta & De Campos Vidal, 1989*; *Shadwick, 1990*) and support high loads (*Franchi et al., 2007*). Moreover, physical exercise can induce morphological and biochemical modifications that alter the biomechanical properties of the collagen bundles, resulting in tissues supporting higher tensile strength (*Vilarta & De Campos Vidal, 1989*). However, changes in training regime also cause connective tissue alterations that in most cases are clearly pathological (*Selvanetti, Cipolla & Puddu, 1997*; *Kader et al., 2002*; *Shwartz, Blitz & Zelzer, 2013*; *Thampatty & Wang, 2018*). The shape of the cells of the fibrocartilage located over the tibia-fibula was also affected in treated juveniles and adults. The flattening of their nuclei (typically rounded) was noticeable in *L. latinasus* and *P. sauvagii* (*Benjamin & Ralphs, 1998*). The flattening of the fibrocartilage nuclei due to intense stimulus was already recorded for menisci cells (*Benjamin & Evans, 1990*), and stressed areas of the tendons (*Carvalho & Felisbino, 1999*).

In the tendinous tissue, the separation of the collagen fibers was similarly accentuated in treated juveniles and adults. Our model shows that the probability to present more changes is remarkable on *R. arenarum*. Tendon alteration has been described as one of the traits characterizing the tendinosis syndrome in mammals (*Selvanetti, Cipolla & Puddu, 1997*; *Kraushaar & Nirschl, 1999*; *Maffulli et al., 2008*; *Kim et al., 2015*; *Thampatty & Wang, 2018*). It has been reported that exercise tends to increase collagen cross-links (*Kannus et al., 1997*). However, when training is extreme, collagen fibers damage, delaying collagen maturation and inhibiting such links (*Kannus et al., 1997*). Interestingly, the collagen cross-link can suffer a similar pattern of degradation and rupture when the tissue

is immobilized (*Selvanetti, Cipolla & Puddu, 1997*), which could explain the similar phenotypes between the tendinous tissue of our treated specimens and those reported by *Abdala & Ponssa (2012)* in the reduced mobility trials (Fig. 6B in *Abdala & Ponssa 2012*).

Collagen fibers arrangement in the menisci was also affected in the experimental groups. The packed arrangement pattern found in adults (both in control and treated specimens) was also found in tadpoles with reduced mobility (*Abdala & Ponssa, 2012*). In humans, a normal meniscus is characterized by packed collagen fibers (*Pauli et al., 2011*), with a high disorganization associated to aging and osteoarthritis (*Pauli et al., 2011*). The histological structure of the menisci appears to be adapted to the weight-bearing function (*Clark & Ogden, 1983*). Our results allow us to suggest that the disarrangement of the collagen fibers of the menisci is not due to injury, but is instead due to young tissue developing a more packed configuration when mechanical stress increases, as it was observed in both treated juvenile and adult specimens. Likewise, in the human menisci, a higher number of cells is common in young tissues (*Senan et al., 2011*), and it decreases with age (*Clark & Ogden, 1983*).

A severe flattening of the hypertrophic chondrocytes was found in treated juveniles of *L. mystacinus*. The observed injuries did not reach the magnitude of those described in some specimens of anurans raised under reduced mobility (*Abdala & Ponssa, 2012*; *Ponssa & Abdala, 2016*). These immobilized frogs presented irregularly-shaped cells, with large lacunae, interlacunar matrix with thin boundaries and flatter than normal, resulting in a characteristic net-like appearance (*Abdala & Ponssa, 2012*). The deformation of cartilaginous cells has been reported as a response to mechanical stress (*Quinn et al., 1998*), but is also related to an unhealthy tissue (*Cook et al., 2004*). Hypertrophic chondrocytes are large and round due to the mineralization of the matrix during endochondral ossification (*Pacifici et al., 1990*). Thus, the observed change in their shape could present interesting consequences for the normal process of endochondral ossification.

The similar characteristics of tissues subjected to overuse trials observed in this study and tissues of reduced-mobility tadpoles (*Kim, Olson & Hall, 2009*; *Abdala & Ponssa, 2012*; among others), suggests that the knee-joint tissues suffer the same kind of alterations under abnormal movement stimuli (i.e., either overuse or disuse of the joints). Even considering the elastic properties of tendons, fibrocartilages and articular cartilages (*Carvalho, 1995*; *Kannus et al., 1997*), limbs overuse or disuse still cause alterations or changes in the biomechanical properties of the connective tissue of the joints (*Järvinen et al., 1997*; *Kannus et al., 1997*; *Cook & Purdam, 2009*) conducting to pathologies such as osteoarthritis or tendinosis. Our results agree with the great amount of evidence associating pathologies to joint overuse, e.g., jumper knee, runner knee, golfer and tennis elbow, among others (*Engebretsen & Bahr, 2007*). Surprisingly, the alteration of the knee tissue was similar in jumper than in walker species, in spite of the profound effect of the sudden and abrupt contact between the long bones epiphyses during the jump. It should be considered, however, the applied mechanical test could have been either too challenging or unsuitable, thus preventing the identification of differences between locomotor modes.

Both knee-joint morphology and locomotion of anurans differ from those of mammals (*Kargo, Nelson & Rome, 2002*). Indeed, in rats, the range of motion is up to 145° (*Nagai et al., 2016*); while in frogs it is about 155° (*Kargo, Nelson & Rome, 2002*). The hind limb bones of anurans do not lie in a single plane throughout the jump, and joint rotations are more prominent than joint extensions (*Gans & Parsons, 1966*), while in mammals the kinematics of the knee consist of flexion-extension movements (*Fischer et al., 2002*). Despite these differences, the effects of overuse and immobility trials over the connective tissues of anurans are similar than those reported in mammals (pigs, rabbits, rats and men) and chicken (*Cook et al., 2004*; *Kannus, 1997*). This implies that anurans could be a good model for studying abnormalities in the development caused by epigenetical stimuli, such as movement alterations (*Ponssa & Abdala, 2016*).

Our study provides a first approximation for the understanding of tissue dynamics of the knee-joints in anurans, taking ontogeny and the different locomotors modes into account. These new data constitute a deeper approximation to the comprehension of the effect of mechanical load in the development and maintenance of knee tissues in tetrapods, which could contribute to the engineering of skeletal tissues (*Nowlan et al., 2010*). Knee alteration and pathologies are caused by an interaction between excessive load (immobilization or excessive movement) (*Ni et al., 2015*) and intrinsic factors, such as genes, age, circulating and local cytokine production, sex, biomechanics and body composition (*Cook & Purdam, 2009*; *Zamli & Sharif, 2011*). Therefore, studies taking these variables into account are necessary for a better understanding of the knee-tissues behavior in tetrapods, and therefore for the treatment and prevention of knee-joint pathologies.

## CONCLUSIONS

Our data showed that anuran knee tissues suffer gradual pathological structural changes when subjected to overuse, especially in adults. The changes observed include disarrangement of the collagen fibers of tendons and fibrocartilage, packaging of the collagen fibers of the menisci and the flattening of the fibrocartilage and the diaphysis cells. Similar effects were found in anurans subjected to immobilization trials, and in joints diseases such as tendinosis and osteoarthritis in mammals. Taken together, these results suggest that the knee tissues of tetrapods tend to react similarly even when subjected to different types of stimuli (i.e., overuse or disuse). This work represents one of the first approaches to the study of knee tissues dynamics when subjected to overuse trials in anurans.

## ACKNOWLEDGEMENTS

We are very grateful to Esteban J. Vera for building the treadmill belt; to Adriana Manzano (CCyTTP, CONICET, Diamante, Argentina) and Gladys Hermida (UBA, Buenos Aires, Argentina), Daniela Miotti and Marcela Hernandez (Fundación Miguel Lillo, Tucuman, Argentina) for their help in the preparation and interpretation of the histological samples. The reviewers' helpful criticism and suggestions improved our work in many ways.

### Funding

This work was supported by AGENCIA NACIONAL DE PROMOCIÓN CIENTÍFICA Y TECNOLÓGICA (Préstamo BID PICT 2015/1618, and PICT 2016-2772), CONICET (PIP N°389). The funders had no role in study design, data collection and analysis, decision to publish, or preparation of the manuscript.

### Grant Disclosures

The following grant information was disclosed by the authors:
AGENCIA NACIONAL DE PROMOCIÓN CIENTÍFICA Y TECNOLÓGICA (Préstamo BID PICT 2015/1618, and PICT 2016-2772), CONICET (PIP N°389).

### Competing Interests

Virginia Abdala is an Academic Editor for PeerJ.

### Author Contributions

- Miriam Corina Vera performed the experiments, analyzed the data, contributed reagents/materials/analysis tools, prepared figures and/or tables, authored or reviewed drafts of the paper, approved the final draft.
- Virginia Abdala conceived and designed the experiments, analyzed the data, contributed reagents/materials/analysis tools, authored or reviewed drafts of the paper, approved the final draft.
- Ezequiel Aráoz analyzed the data, approved the final draft.
- María Laura Ponssa conceived and designed the experiments, analyzed the data, contributed reagents/materials/analysis tools, authored or reviewed drafts of the paper, approved the final draft.

### Animal Ethics

The following information was supplied relating to ethical approvals (i.e., approving body and any reference numbers):

Experiments were approved by the Ethical Committee of Facultad de Medicina, Universidad Nacional de Tucumán (Res. No. 81962-2014).

### Field Study Permissions

The following information was supplied relating to field study approvals (i.e., approving body and any reference numbers):

Field experiments were approved by Dirección Pcial. de Fauna, Tucumán (Res. No.13-16), Dirección Pcial. de Fauna, Salta (Res. No. 0308/14) and Dirección Pcial. de Fauna, Jujuy (Res. No. 21/2012).

### Data Availability

The raw data are provided in the Supplemental File.

## Supplemental Information

Supplemental information for this article can be found online at http://dx.doi.org/10.7717/peerj.5546#supplemental-information.

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
