# Peer review of "Movement and joints: effects of overuse on anuran knee tissues"

_PeerJ, doi:10.7717/peerj.5546_

## Round 0.1 · original submission · Major Revisions

We obtained very mixed reviews for this study. One reviewer was effusively positive but very brief. Another had substantial criticisms but also saw some potential value to the study. A third had extreme concerns about the study on several grounds. The study clearly was a major undertaking in terms of effort, but the major concerns raised include (1) if the scoring system used is sufficient, (2) if the statistical methods and experimental controls are sufficient, and (3) if the study is ethical in terms of animal welfare. More evidence than a "yes" answer to these 3 points will be needed to bring the study to an acceptable level, but there might be flexibility. Item #3 is a major concern and in itself brings the study very close to outright rejection. However it is possible that there was miscommunication and so the authors may elect to revise the paper with an itemized rebuttal to all reviewers' points. This may still result in rejection if the reviewers are not won over, especially on item #3. I am sorry for this difficult news but must uphold the journal's criteria. If the authors choose to redesign the study and resubmit a new version later after withdrawing this one, or submit elsewhere, we understand, but we hope that the reviewers' points, which are reasonable, are used to revise any future version of the manuscript. I wish you good fortune in revising this study as I can see that the motivation was genuinely to reveal new insights into anuran mechanobiology, which is indeed poorly known.

Reviewer 1 ·

Basic reporting

Overall, the paper met the primary standards of being self-contained, having proper tests of the hypotheses and relevant results, and sufficient background information. The author's figures were very well done, and I would like to particularly single out Figure 5 as the most useful methods figure I've seen in a while. Overall, the text was quite readable, but there were some odd word choices and unclear sentences (particularly line 139-140); I would recommend a native English speaker giving it a look over and some suggestions.

Experimental design

The experimental design was largely satisfactory, except for the tissue alterations score, which had two significant issues.

First, the coarse 0-1-2 raking doesn't leave a lot of room for subtlety. The scales these were based on (Maffulli et al 2008 and Pauli et al 2011) have four categories, not three, so why were the number of categories reduced?

Second, no statistical analysis methods were included for this data, not even Chi-squared tests. One cannot simply take the distributions of categorical variables at face value, anymore than one can for continuous variables.

Additionally, no method for adjusting for multiple tests was included.

Finally, although multiple species were used, there was no consideration of differences between species or locomotor style given, beyond cursory mention in one table, and no statistical framework for evaluating this was provided. Why were these data pooled? At the very least, jumpers vs walkers should be separated, given the dramatic difference in the loading levels presumably experienced by the joints.

Validity of the findings

The fundamental problem with this paper is that the tissue alteration scores were not statistically evaluated. As stated in the prior section, just as with continuous variables, you cannot assume differences in categorical variable distributions aren't the product of random chance and sampling error, especially with limited sample sizes. With a few exceptions, the differences between control and treatment groups was small, only a few samples difference, as shown in Figure 6. Perhaps these are real, but without statistical testing, there is no was to be sure. Almost the entire discussion rests upon these results, making it all of questionable validity. These results need to be tested in a statistically rigorous way which accounts for the categorical nature of the response variable and the multivariate nature of the evaluation.

The articular cartilage cell quantification runs into another problem of significance: in spite of no significant differences being found between treatment and control in either bone's articular cartilage, the authors insist on discussing these non-significant differences in the results (lines 282-286) and discussion (343-369). Examination of Figure 8 does not reveal any visible trend matching the author's claims, regardless of statistics.

Per this journal's mission, negative and inconclusive results are not viewed unfavorably, but the statistical tests must be conducted in the first place (tissue alteration scores) and acknowledged as negative when the statistics show it (articular cartilage).

Additional comments

Lines 91-93 are not up to date. Nauwelaerts & Aerts 2003 did not actually quantify long-axis rotation, and while Kargo, Nelson & Rome 2002 predicted it based on a simulation, no long axis rotation of any limb bone was observed in Astley, H. C., and Roberts, T. J. (2014) The mechanics of elastic loading and recoil in anuran jumping. Journal of Experimental Biology, 217, 4372-4378

In line 91, I would add the following citation, which shows the highest peak jump forces known in anurans: Work and power output in the hindlimb muscles of Cuban tree frogs Osteopilus septentrionalis during jumping. Peplowski M, Marsh R. The Journal of experimental biology. 1997 vol: 200 (Pt 22) pp: 2861-70

I would add "anuran knee tissues" to the title for clarity.

The supplementary video was entirely superfluous, and can be omitted. For citation on the largely sit-and-wait lifestyle of anurans, see: Movement patterns in leiopelmatid frogs: Insights into the locomotor repertoire of basal anurans. Reilly S, Essner R, Wren S, Easton L, Bishop P. Behavioural Processes 2015 vol: 121 pp: 43-53.

Reviewer 2 ·

Basic reporting

The article “Movement and joints: Pathological effects of overuse on knee tissues of frogs (Reviewing Manuscript 21173v1)” represents a pioneer study about the understanding of the dynamics of anuran knee tissues during the ontogeny and in relation to locomotion. It was written using clear and professional standards of the scientific language. The English wording is very good. The article included sufficient background in relation to theme and showed a good presentation of the field of knowledge. The structure of the article is organized in the traditional format of scientific manuscript sections. The figures represent the content of the article and they are appropriately described and labeled. All data has been made available in accordance with PeerJ Data Sharing policy.

Experimental design

The research question is relevant and clearly defined. The investigation method was described with a high technical standard and conducted in conformity with the ethical guidelines of the animal experimentation.

Validity of the findings

The data are robust and they are related with the conclusions. These were connected to the original questions and supported by the results.

Additional comments

No comments.

Reviewer 3 ·

Basic reporting

The reporting is clear and the English is mostly good. Its purpose is however, not clear and the conclusions rather sketchy at best.

The article fails to include sufficient introduction and background to demonstrate how the work fits into the broader field of knowledge. Is there work intended to identify physiological homeostatic changes in tissue architecture induced by an extreme form of exercise or instead to detail the pathology that this kind of challenge produces?

The authors provide Figures showing the whole joint in ‘normal’ anurans but fail to provide similar for those that have been exercised.

Extensive reference is required to prior work by the same authors. Without this their current submission is very difficult to assimilate into a body of work. In particular, the last sentences of the introduction are completely dependent on their prior work, which is not discussed.

Experimental design

Their research question is at best cumbersome to negotiate; it is not particularly well defined or meaningful. Its failure seems to centre around whether there were any merits in studying multiple species, the questionable ‘end points’ the authors select to observe/measure and the methods employed to select and to ‘exercise’ their animals.

The submission does not clearly define a relevant and meaningful research question and hence the knowledge gap being investigated is too ill-defined.

The study design is not rigorous enough, with inappropriately calibrated protocols and end points that do not provide any specific new insights. The investigation is neither performed to a high technical or, seemingly, ethical standard.

The extent of the exercise regimen seems to be based mostly on the basis of the animals becoming exhausted – refusing to complete the task – there are many factors that can contribute to this ‘decision’ and it seems to this reviewer that some quantifiable parameter needs to be used to calibrate this system for the proposed use across animals of different age and species.

Methods: How are juveniles and the attainment of adulthood defined; how long after adult status is reached was it permitted until an animal was deemed too old for inclusion.

Methods: How long were they houses in individual terrariums prior to the start of the imposed treadmill exercise regimen? Was this short enough for there to be no evidence of structural adaptation in the housed animals – adult and juvenile?

Lines 133-4: this implies that there was no growth is the size of the adults or of the juveniles during the course of these studies. Is it surprising that there was no growth in juvenile animals?

Lines 140-1: indicates that there is little movement in these species. Does this make the author’s choice to study adaptation to exercise in these species questionable?

Line 151-3: The exercise regimen seems to be rather excessive in my opinion; fatiguing animals twice daily for 59 days would raise serious welfare issues and, at the very least, require that another group, utilising the same ‘energy’ but without the treadmill endurance exercise, be sacrificed as a control.

Line 157-8: this appears to suggest that a single speed was chosen. This seems rather inappropriate given that animals at different stages of maturity and different species are being exercised. Surely such a system needs some form of calibration in order to make this use of different stages/species meaningful.

Line 196: These methods are using a very ill-defined scoring system that has never been used, to my knowledge, before. Its justification and validation need to be performed in order to give these experiments any meaningful experimental value for other researchers.

Structural disorganisation can be taken to mean very many things. This kind of classification is not helpful.

Validity of the findings

The relevance of the study design and the end points selected for measurement are not obvious.

Additional comments

It is not clear upon which basis the authors have been led to hypothesise that ‘adults will experience deeper morphological changes than juveniles’.

The authors should explain why there is any benefit from examining responses in small numbers of several anuran species rather than larger numbers of one species.

The authors justify the choice of anurans based on the larger joint stresses. What is the evidence for this assertion?

Line 64-66: phrasing ‘does not have time to repair itself’ supposes these are responses to injury, rather than some form of remodelling, which is considered a homeostatic adaptive response.

Lines 104-107: The question to which this work is addressed is not clear.

Lines 107-112: ‘a structural disorganisation process.’ Is this pathological? If the authors are supposing that adult tissues cannot adapt then this is incorrect. It is well established that periods of rapid growth are conducive to adaptation to applied loading regimes. If this is testing the same idea, then it does so rather inelegantly.

Results
Lines 217-233: This reviewer finds it rather difficult to ascertain the value this data provides.
The authors make use of the term laxer as though they have measured collagen fibre resilience/elasticity based on their compactness in histology. No such measurements are made – they are simply commenting on histology.

---

## Round 0.2 · Major Revisions

One reviewer remains pleased with the paper, one very critical reviewer has been unable to review this time but was replaced by a reviewer who feels that revisions have been adequate, and a reviewer who previously was at least moderately critical now is even more critical. My judgement is that one more round of revision is needed to better address their points. I am sympathetic that obtaining more frogs at this point is not feasible but the reviewer (#1) makes good suggestions for doing better statistics with the data already obtained, and making the existing methods regarding statistics clearer. Without such amendments we cannot accept the study. However, I am optimistic that with some extra effort (I highly recommend consulting with a statistician) the paper can be accepted.

Reviewer #4 (new reviewer) does make some points about the possible ambiguity of the results -- "On the other hand, the fact that there was no difference between jumpers and walkers, who most likely represent two different mechanical strategies, might suggest that the mechanical test that was applied was either to challenging or unsuitable, thus preventing the identification of differences between species." Please address this point in the revised MS (i.e. in the paper, not just response to reviewers.

I will check the Response and revised MS and make a final decision once the resubmission is made. I look forward to seeing the amended paper, and thank the authors for their patience through a challenging review process. I also apologize for delays- the final revision will be processed more quickly.

Reviewer 1 ·

Basic reporting

The paper meets basic reporting requirements as above, though there remain some oddities of grammar and word choice which can make for puzzling sentences. Figures and tables are good, though the primary subject of the statistical test, the combined tissue alteration score, is not given in any figure or table (though the components are in Figure 7).

Experimental design

The justification of the reduced number of categories in the categorization is uninformative and poor. The text addition simply says the authors “adapted it to be used in comparative biology”, without further justification or explanation of why such adaptation was necessary or beneficial. The review response was more elaborate but no more informative. The authors state it was for accessibility, but this is a poor reason to choose an experimental metric – the objective of the data is to accurately represent reality, and it is the most representative data which should be used; aids to understanding should be confined to the discussion. The underlying variables are fundamentally continuous, although their responses to exercise are correlated, so why not use either the raw variables or a PCA thereof? Converting continuous data into categorical data erases information, and choices of cutoffs are crucial. But even if we accept that the categorical framework of Maffulli and Pauli is accurate and statistically valid, consolidating four categories into three reduces statistical degrees of freedom and erases variation. Ultimately, the question is “why not just use the underlying continuous data, kept in a continuous form?” must be answered, and “ease of reading” is not sufficient justification for the information loss. Similarly, if categorical data should be used, “why were the number of categories and their cutoffs chosen?” must be answered. If the authors simply used the Maffulli and Pauli categorization intact, they could “pass the buck” on both, but altering that categorization requires justification beyond simply accessibility. If Maffulli and Pauli relied upon more variables, why didn’t you quantify all of their variables?

The statistical evaluations have improved, simply by virtue of existing, but are still insufficient. The authors claim that the use of wild animals results in low sample size (something I’m not unsympathetic to), yet the authors note in their ethical justifications that these species are not rare and reproduce in large numbers. Why not simply go catch more frogs, particularly during the next breeding season, since they are all native to the region? This is essential, because you have multiple independent variables – exercise vs. not, juvenile vs. adult, and walker/nonwalker (with species being a nested factor within this). Multivariate methods are never used, and instead each independent variable is tested separately, with no adjustment for multiple tests or possible confounding or co-variation.

Validity of the findings

The current statistical tests for “tissue alteration scores”, which seem (the authors do not include sufficient detail in the methods to be sure) to be the results of two tests, globally pooling all adults and all juveniles and testing each dataset for exercise vs. not, without accounting for species or locomotor mode (the effects of which were tested separately in a manner not sufficiently detailed in the paper). Because species and walker/not are not included in these tests (but rather tested separately later), no consideration is given to whether these factors could influence the outcome, especially given the total lack of juveniles of Melanophryniscus and minimal numbers in Phyllomedusa. It should be noted that the combined “tissue alteration scores” which are tested are not shown in any graph (Fig. 7 shows the components, but not the whole). The juveniles shows a significant difference (if we accept the authors’ reduced categorization), and adults do not, but no adjustment for repeated testing (e.g. Bonferroni correction) is used. The authors note the lack of significant effects for adults, but then proceed on a statistical fishing expedition (through methods unspecified) to post-hoc justify the exclusion of one of their variables. Furthermore, none of the asserted differences in Figure 7 were tested for statistical significance; instead, the authors seem to have assumed that if the overall result for combined score was significant, all of the underlying variables must also be significant, which is unfounded to say the least.

Additional comments

No further comments

Reviewer 2 ·

Basic reporting

No comment.

Experimental design

No comment.

Validity of the findings

No comment.

Additional comments

The article meets conditions to be accepted in the presented format.

Reviewer 4 ·

Basic reporting

The connection between movement and joint formation is well-established. In this work, the effect of extreme exercise on joint morphology was studied using comparative analysis between wild animals of five species of anurans that were trained to run on a treadmill. The findings are that intense movements lead to changes in morphology, that young joints are more tolerant to stress than those of adults and, finally, that there is no difference between jumpers and walkers.
Overall, this is a limited study as are the conclusions that can be drawn from it. One interesting finding is that extensive usage can lead to morphological changes. On the other hand, the fact that there was no difference between jumpers and walkers, who most likely represent two different mechanical strategies, might suggest that the mechanical test that was applied was either to challenging or unsuitable, thus preventing the identification of differences between species.
Regarding the revision, it appears that the authors made an honest effort to address most of the comments of reviewer 1, but largely objected the comments of reviewer 3, which were indeed very negative and very difficult to reconcile with the author's approach. Because I got involved only during the revision stage, it is very difficult for me to decide categorically between the two sets of arguments.

Experimental design

The connection between movement and joint formation is well-established. In this work, the effect of extreme exercise on joint morphology was studied using comparative analysis between wild animals of five species of anurans that were trained to run on a treadmill. The findings are that intense movements lead to changes in morphology, that young joints are more tolerant to stress than those of adults and, finally, that there is no difference between jumpers and walkers.
Overall, this is a limited study as are the conclusions that can be drawn from it. One interesting finding is that extensive usage can lead to morphological changes. On the other hand, the fact that there was no difference between jumpers and walkers, who most likely represent two different mechanical strategies, might suggest that the mechanical test that was applied was either to challenging or unsuitable, thus preventing the identification of differences between species.
Regarding the revision, it appears that the authors made an honest effort to address most of the comments of reviewer 1, but largely objected the comments of reviewer 3, which were indeed very negative and very difficult to reconcile with the author's approach. Because I got involved only during the revision stage, it is very difficult for me to decide categorically between the two sets of arguments.

Validity of the findings

The connection between movement and joint formation is well-established. In this work, the effect of extreme exercise on joint morphology was studied using comparative analysis between wild animals of five species of anurans that were trained to run on a treadmill. The findings are that intense movements lead to changes in morphology, that young joints are more tolerant to stress than those of adults and, finally, that there is no difference between jumpers and walkers.
Overall, this is a limited study as are the conclusions that can be drawn from it. One interesting finding is that extensive usage can lead to morphological changes. On the other hand, the fact that there was no difference between jumpers and walkers, who most likely represent two different mechanical strategies, might suggest that the mechanical test that was applied was either to challenging or unsuitable, thus preventing the identification of differences between species.
Regarding the revision, it appears that the authors made an honest effort to address most of the comments of reviewer 1, but largely objected the comments of reviewer 3, which were indeed very negative and very difficult to reconcile with the author's approach. Because I got involved only during the revision stage, it is very difficult for me to decide categorically between the two sets of arguments.

Additional comments

The connection between movement and joint formation is well-established. In this work, the effect of extreme exercise on joint morphology was studied using comparative analysis between wild animals of five species of anurans that were trained to run on a treadmill. The findings are that intense movements lead to changes in morphology, that young joints are more tolerant to stress than those of adults and, finally, that there is no difference between jumpers and walkers.
Overall, this is a limited study as are the conclusions that can be drawn from it. One interesting finding is that extensive usage can lead to morphological changes. On the other hand, the fact that there was no difference between jumpers and walkers, who most likely represent two different mechanical strategies, might suggest that the mechanical test that was applied was either to challenging or unsuitable, thus preventing the identification of differences between species.
Regarding the revision, it appears that the authors made an honest effort to address most of the comments of reviewer 1, but largely objected the comments of reviewer 3, which were indeed very negative and very difficult to reconcile with the author's approach. Because I got involved only during the revision stage, it is very difficult for me to decide categorically between the two sets of arguments.

---

## Round 0.3 · accepted · Accept

I am satisfied by the revisions, which seem reasonable to me. I do not feel that further review is warranted at this stage.

However, given that the paper has had some criticism on the grounds of the statistics, and this is an Open Science journal, I strongly encourage you to put all of your data (i.e. those data points that the stats were applied to) into the supp info for the paper, or in an online repository with a link provided in the paper. That would serve you and the scientific community best.

Thank you for your patience and dedication, and congratulations!

#